# Investigation of the Damage Phenomenology with Dependence on the Macroporosity and Microporosity of Porous Freeze Foams

**DOI:** 10.3390/ma16062484

**Published:** 2023-03-21

**Authors:** Johanna Maier, David Werner, Vinzenz Geske, Thomas Behnisch, Mathias Ahlhelm, Tassilo Moritz, Alexander Michaelis, Maik Gude

**Affiliations:** 1TU Dresden, Institute of Lightweight Engineering and Polymer Technology, Holbeinstraße 3, 01307 Dresden, Germany; 2Fraunhofer Institute for Ceramic Technologies and Systems, IKTS, Winterbergstraße 28, 01277 Dresden, Germany; 3Fraunhofer Institute for Ceramic Technologies and Systems, IKTS, Maria-Reiche-Str. 2, 01109 Dresden, Germany

**Keywords:** ceramic foam, compressive strength, porous ceramics, non-destructive testing

## Abstract

Freeze Foams are cellular, ceramic structures with hierarchical pore structures that are manufactured using the direct foaming process. By tailoring their morphology and strength, these foam structures are able to cover a wide range of application. Earlier works identified that pore-forming influencing factors (water and air content, suspension temperature, as well as pressure reduction rate) dictate the constitution on a macroscopic and microscopic scale. Therefore, the ability to manufacture foams whose properties align with the component requirements would be an important step in advancing towards a widespread application of these promising materials. With this goal in mind, the correlation between the pore-forming influencing factors and the resulting mechanical properties was quantified. Foams with independently adjustable porosities were produced at the micro and macro scales and evaluated according to their material failure behavior under compressive loads. As a result, foams with determined macroporosities between 38 and 62%, microporosities between 25 and 42%, and compression strengths between 1 and 7 MPa with different material failure characteristics were manufactured and systematically investigated.

## 1. Introduction

Hierarchical, porous ceramics can have remarkable multifunctionality [1] due to their different properties, such as the pore size distribution or connectivity, which offer a wide range of applications, such as filters, catalysts [2,3,4], or bone replacement material [5]. Meanwhile, the mechanical properties of these foams are dependent on their macroscopic and microscopic constitution. The average sizes of the pores, as well as their size distribution and the fraction of closed and open porosity impact both the stress–strain curve and structural strength (e.g., compression strength) significantly [6]. The latter is especially susceptible to changes in the microstructure of cell struts. By adjusting the foam structure in terms of its macroporosity and microporosity during the manufacturing process, it is therefore possible to adjust the mechanical properties.

Hierarchical foams are characterized by a bimodal distribution of microscopic and macroscopic pore sizes, as well as high interconnectivity [7] paired with good mechanical properties [8]. A high degree of open porosity in bioceramic foams results in a high specific surface area, allowing biological tissue to grow on and into the foam structure [5]. To manufacture such foams, several production technologies are available, e.g., partial sintering, freeze casting, sacrificial fugitives, replica templates, direct foaming, as well as certain 3D printing techniques [3,9,10,11,12,13].

In this work, the so-called freeze foaming process was used to produce porous foams. The resulting cell structure was generated via a pressure-induced and pressure-controlled inflation of a suspension made from water and a material powder (e.g., ceramic powder) with subsequent freeze-drying [14]. Freeze Foams are characterized by a hierarchical porous structure, exhibiting a macroporosity with foam cell sizes between 10 and 1000 μm and a microporosity created by sublimating ice crystals with pore sizes between 0.5 and 10 μm. An earlier investigation identified the pore-forming influencing factors for the foaming process (water and air content of the suspension, suspension temperature, and pressure reduction rate) [15]. An adapted process control [16], as well as the choice of a particular suspension composition allow the tailoring of the macrostructure and microstructure. As the ability to tune the macroporosity and microporosity allows defining both the strength and the stress–strain behavior of foamed structures, it is necessary to adjust the mechanical and structural properties. Hence, this paper aimed to derive and quantify the correlation between the pore-forming influencing factors and the mechanical and structural properties, as well as to characterize the material characteristics of sintered samples exhibiting tunable combinations of macrostructure and microstructures. The damage phenomenology of Freeze Foams was investigated by means of acoustic emission analysis based on the pore-forming influencing factors and correlated with the microstructure and macrostructure. The validation was performed by using X-rays, field emission scanning electron microscopy (FESEM), and mercury porosimetry.

## 2. Materials and Methods

### 2.1. Material

For this model suspension, hydroxyapatite (HAp) (Merck KGaA, Darmstadt, Germany; BET = 70 m^2^/g, d50 = 2.64 μm) was chosen as the bioceramic powder. Apart from HAp, the suspension consisted of 4.6% dispersant (Dolapix CE64, Co. Zschimmer & Schwarz Mohsdorf GmbH & Co. KG, Burgstädt, Germany), 1.3% binder (Mowiol 20–98, Merck KGaA, Darmstadt, Germany), 3.1% thickener (Tafigel AP15, Co. Münzing Chemie GmbH, Heilbronn, Germany), and 2% alkaline substance (2-amino-2-methylpropanol (AMP), Merck KGaA, Darmstadt, Germany) in relation to the powder mass. The suspension was prepared with a centrifugal vacuum mixer. Freeze foaming was conducted within the freeze dryer ALPHA 2-4 LSCPLUS (Co. Martin Christ Gefriertrocknungsanlagen GmbH, Osterode, Germany). The resulting freeze-dried foams were debindered to remove the organic material and subsequently sintered at 1280 °C for 1 h. The whole process is described in detail in [15]. Table 1 shows the chosen parameter configuration for the characterization of the microscopic and macroscopic foam structure, as well as the resulting compression strengths. Different variations of the water content (W) in weight-%, degassing (D) of the suspension before evacuation in minutes, suspension temperature (T) in °C, and pressure reduction time (P) in minutes (from ambient pressure to 2 mbar) were examined during the tests.

### 2.2. Methods

Macrostructure analysis including pore the size distribution was performed using an X-ray computed tomography (CT) device (CT Compact, Co. Procon-X-Ray. Sarstedt, Germany). The scanning parameters were set to an acceleration voltage of 110 kV and a beam current of 100 μA. The resulting voxel size was 28.3 μm. The analysis of the 3D data was conducted using VGStudio Max 3.0 (Volume Graphics GmbH, Heidelberg, Germany).

The morphology of the strut pores (microstructure) was analyzed with the help of FESEM (ULTRA 55, Co. Carl Zeiss, Oberkochen, Germany). Specimens were cut using a diamond cutting disc and then prepared for ceramography. A backscattered electron detector was used to investigate the material contrast between the HAp and resin-filled pores. The pore size distribution of the strut pores was examined with a mercury porosimetry device (AUTOPORE V, Co. Micromeritics Instrument Corporation, Norcross, GA, USA). Due to the nature of this method, only open pores between 0.004 and 400 μm can be detected. The determination of the strut porosities was performed as demonstrated in [17].
(1)$$StP=1-\frac{m_{foam}/V_{strut}}{\rho_{struct}}$$

The material volume of the scanned foams represents the strut volume $V_{strut}$ including the strut pores. The quotient of foam mass $m_{foam}$ and $V_{strut}$ is therefore equal to the geometrical density of the foam struts. The bulk density was measured with helium pycnometry after sintering.

Subsequently, sintered samples were compression tested on the universal testing machine Zwick Roell 2.5 kN (Zwick GmbH, Ulm, Germany) with a strain rate of 1 mm/min after the top and bottom sides of the rectangular samples were cut to be parallel and the dimensions were measured.

To characterize the damage phenomenology and correlate it with the measured stress–displacement curves, acoustic emission analysis was performed. Triggered by damage phenomena and the subsequent release of elastic energy inside the specimen, the summation of sound-emitting events was recorded. The tests were conducted using the modular acoustic emission system AMSY-5 (Vallen Systeme, Wolfratshausen, Germany). The acoustic sensors used were highly sensitive VS-150-M piezo sensors, also manufactured by Vallen Systeme. During testing, both the force–displacement curve, as well as acoustic noise events (also called hits) were recorded and correlated in a time-synchronous manner with the mechanical structure behavior. The hit event indicator only regards signals above a pre-defined dB threshold. Consequently, the hit sum is the number of incidents during testing in which the signals exceeded this threshold. Looking at the surges and plateaus allows identifying the time intervals with exceptionally high or low damage activity.

## 3. Results

### 3.1. Macrostructure—Foam Cells

Both the macroscopic porosity ($P_{M}$) and the pore size distribution were examined to analyze the macrostructure (Figure 1 and Figure 2). The highest measured porosity ($P_{M3}$ = 62.48% and $P_{M2}$ = 62.82%) was observed in samples with 34 m.% water (W34), a 23 °C suspension temperature (T23), and a 0 min degassing time (D0), while the influence of the pressure reduction rate can be neglected. On the other hand, 3-minute degassed samples (D3) foamed at a 5 °C suspension temperature (T5) exhibited the lowest porosities ($P_{M4}$ = 38.49%). It was evident that the macrostructure was most affected by the degassing time D (Sample 4) caused by the air introduced into the suspension during mixing. During the pressure reduction, the air expands, generating macroscopic pores (foam cells). Hence, a higher air content (a shorter degassing time) will result in higher macroscopic porosity. Temperature exhibited the second-biggest impact on the porosity: higher suspension temperatures led to a lower viscosity and a higher vapor partial pressure during the evacuation, promoting the growth of bubbles and, consequently, a higher macroscopic porosity. In contrast, a too high water content (W48) reduced the macroscopic porosity (Sample 5), as this can lead to cell wall destruction and a subsequent collapse of the foam structure.

The influence of the manufacturing parameters on the pore size distribution is analogous to that of porosity. The largest macroscopic pores developed from degassed samples (Sample 4—d50 = 648 μm). Due to the low amount of solved or enclosed air bubbles leading to much less interfering effects such as Ostwald ripening or coalescence, they were able to grow during pressure reduction with little interference from neighboring bubbles. Here, the air content exhibited the highest influence on the development of the macroscopic pores. The smallest average pore sizes were observed in T5 samples (Sample 1—d50 = 280 μm) because of their high viscosity. On the other hand, decreasing the viscosity due to a higher water content (W48) resulted in larger macroscopic pores (Sample 5—d50 = 479 μm) [15,16]. The influence of the pressure reduction rate on the pore size distribution was also negligible.

### 3.2. Microstructure—Strut Pores

To analyze the microstructure of freeze-foamed samples, the porosity and pore size distribution were examined. The water content was identified to have the highest influence on the strut pore formation. Samples with a high water content (W48) exhibited the highest porosity (${\mathrm{StP}}_{5}$ = 42%), while the lowest porosity of 25.1% was observed in Sample 2 (W34D0T23P2). Increasing the suspension temperature led to increased evaporation during the foaming process, resulting in significantly reduced microscopic porosity. With the suspension temperature exhibiting the second-highest influence on the microscopic porosity and pressure reduction rate without any significant influence, reducing the air content resulted only in a small increase in the microporosity.

The pore size distribution was characterized by three distinct peaks [15] (Figure 3), whose mean values are shown in Table 2. Peak 1 (0.01– 1 μm) only existed in samples with a low water content (W34). This was likely caused by micro-cracks connecting adjacent strut pores. Peak 2 encompassed the actual microscopic pores and was divided into two separate regions: Region 1 was caused by sinter pores, which could be found in all samples. The pores in Region 2 were lamellar pores (>3 μm) created by freezing and the resulting formation and growth of ice crystals. These types of pores are promoted by either high water content (Sample 5) due to excess water [15] or lower pressure reduction rates (Sample 3), which slows down freezing. The lowest pore sizes were observed in degassed samples (Sample 4). Only pores with a size of 0.004 to 400 μm can be analyzed using Hg porosimetry. As a result, Peak 3 (displayed in grey) was cut off at the upper limit. However, pores of that size were already covered by observation concerning the macroscopic pores in Figure 1 (Section 3.1).

The mean values of the pore sizes from the microscopic and macroscopic pores are summarized in the following Table 2.

Figure 4 depicts the FESEM pictures of all samples, as well as the corresponding strut porosities (StPs). The areas shown in those pictures are not representative of the whole sample structure.

The high influence of the water content is clearly visible in these images, as increasing the water content also increased the strut porosity. Shown also are the lamellar pores created by freezing (>3 μm—Region 2), which were observed in Sample 3 and Sample 5. These types were rarer in Sample 3 and very common in Sample 5. In Samples 1, 2, and 4, no sign of structures caused by freezing could be observed as the higher concentration of ceramic particles in the suspension inhibited the growth of the ice crystals [18].

### 3.3. Compressive Strength and Damage Phenomenology

The compression strength of the specimens with five different parameter sets was determined using quasi-static compression tests. The results of these tests are illustrated in Figure 5. The maximum fracture force was dependent on the sample configuration and the macroscopic and microscopic porosity, respectively. Sample 1 (W34D0T5P2) exhibited the highest strength and low variance (standard deviation DV—0.38). On the other hand, the lowest strengths were measured in samples with a high water content (Sample 5—W48D0T23P2). The largest statistical spread (DV—1.81) in maximum fracture forces was observed in degassed samples (Sample 4—W34D3T23P6).

Figure 6 shows a representative stress–displacement curve for each configuration to characterize the deformation and fracture behavior. Three different modes of failure were identified and are schematically represented in Figure 7.

A—**High and well-defined force maximum due to partly closed macroscopic pores with low porosity and closed microscopic pores**: Sample 1 (W34 and T5) exhibited the highest compression strengths with a well-defined and high maximum force and low deformation. Because of their high suspension temperatures and low water content, these samples showed the smallest macroscopic pores ( 280 μm). Another reason for this high compression strength was the shape of the strut pores. Round, independent pores can hinder and stop the propagation of cracks inside the struts. Additionally, these samples exhibited both a low macroporosity and the lowest geometrical porosity (GP 69.1%) due to the reduced suspension temperature and, therefore, the highest strengths.

B—**Medium force maximum because of interconnected macroscopic porosity with a high total porosity and closed and a low number of micropores**: Samples 2 and 3 showed medium force maxima and low deformation, as well as the lowest microporosity due to the high suspension temperature and low water content. At the same time, they exhibited the highest macroscopic porosity. As Sample 3 only had a few areas with lamellar structures created by freezing, their influence was small. Samples with reduced air content (Sample 4) showed a significantly less homogeneous constitution and a very wide distribution of the compression strength (2 to 7 MPa), which are caused by the very large macropores (Peak 3— 648 μm). The reproducibility of these samples is therefore hardly possible, and the statements can only be applied to the samples investigated here. On the other hand, the influence of the pressure reduction rate on the structural stability was negligible.

C—**Almost constant, low resistance to deformation due to interconnected macroscopic and microscopic pores and high porosities**: Sample 5 with its increased water content exhibited low resistance to deformation during the whole test. These samples showed by far the highest strut porosities, which also demonstrated the high impact of the micropores on the structural stability. The high number of lamellar pore structures created by freezing (Figure 3) additionally reduced the compression strength of the foams. This was caused by an easier crack propagation and the fact that it was easier for lamellae to glide off each other, as described in [18].

To describe the damage phenomenology, characteristic foams were examined using acoustic emission analysis. Different damage processes have specific acoustic emission intensities. Therefore, it is possible to identify the characteristic phases of sample damage over the course of mechanical testing. Four different phases were determined: I—linear elastic section; II—initial damage section; III—transition section; IV—post-fracture section. Figure 8 depicts an exemplary curve (accumulated hits over time) for Failure Type A. During the linear elastic material response (I), no acoustic events were recorded, resulting in a slope of 0. The initial damage section (II) was characterized by the highest slope, indicating a quick progress of the damage. After that, the damage lessened and the slope became less steep (III—transition section) before post-fracture section (IV) was reached, which showed a gradual increase in the hit number, resulting in a decreased ability of the foam to absorb any more energy until the complete failure of the material occurred. For each of the three identified failure types, damage phases I–IV were individually pronounced. Type A showed a clear initial damage section and a distinct transition section. On the other hand, Failure Type C exhibited no concise initial damage section. Instead, damage phases III and IV had higher significance. Type B showed a similar course as A, with a less pronounced initial damage section.

## 4. Discussion

Pore size and pore size distribution, as well as the ratio of open and closed porosity significantly influence both the stress–strain behavior and the structural strength (e.g., compression strength) of Freeze Foams. The conducted research showed a significant correlation between the macroscopic and microscopic structure of Freeze Foams and the type and magnitude of structural failure. Depending on the manufacturing parameters, crucial structure–property relationships were identified.

The highest impact on the macroporosity and pore size during our tests was demonstrated by Factors D and T. Meanwhile, degassed samples exhibited the lowest porosities (Sample 4). A major factor for the formation and growth of macropores during pressure reduction was the entrapped air after suspension preparation. Since there were fewer bubbles in the degassed suspensions, those bubbles were able to expand unhindered, leading to fewer, but larger pores in the finished foam. Due to the influence of the suspension temperature on the vapor partial pressure, a higher suspension temperature (T23) resulted in larger macroporosities, while the T5 samples exhibited the smallest macropores. Furthermore, the temperature-dependent viscosity of the suspension also influences the macropores. Overall, the highest macroporosities were found in the T23 W34 samples and the lowest in the D3 samples (degassed). However, the results of the latter were characterized by a wide spread due to a lack of reproducibility. In any case, no significant influence of the pressure reduction rate on the macropores could be found.

W and T were identified as the most-influential factors on the microscopic porosity. Increasing the water content led to a significant increase in microporosity (Sample 5). At the same time, increasing the temperature of the suspension resulted in a reduction of the microporosity due to higher evaporation rates during pressure reduction. Using Hg porosimetry and subsequent FESEM examination, the formation of lamellar pores created by freezing water (Peak 2.2) were found in samples with a higher water content (W48). They occurred in addition to sinter pores (Peak 2.1) and can impair the mechanical strength of the resulting freeze foam significantly. Hence, the water content must be considered as highly influential.

As a result, three modes of failure (A, B, and C) of the structural stability could be identified, whose occurrence depended on the microporosities and macroporosities exhibited by the different samples:A—High and well-defined force maximum due to partly closed macroscopic pores with low porosity and closed microscopic pores, which was observed mostly in samples with W34 and T5.B—Medium force maximum because of interconnected macroscopic porosity with a high total porosity and closed and a low number of micropores that was the result of elevated temperatures (T23) and low water content.C—Almost constant, low resistance to deformation due to interconnected macroscopic and microscopic pores and high porosities that was caused by high water content (W48) and the resulting lamellar pores due to freezing.

Acoustic emission analysis allowed for the description of the damage phenomenology and the identification of four damage phases. The T5 samples exhibited a distinct initial damage section, while samples with higher water content possessed no clearly identifiable initial damage section.

## 5. Conclusions

This analysis demonstrated that the modification of the manufacturing process parameters allowed an independent adjustment of the macrostructure and microstructure of Freeze Foams. This enabled the targeted manipulation of both the mechanical properties and the types of structural failure and opens the door to establish Freeze Foams for additional targeted use applications. For example, a lamellar microscopic porosity can be useful for applications that require a high accessible surface area, e.g., catalyst supports or adsorber materials. It is also known to be beneficial for applications as bone replacement materials, because nutrient delivery, cell proliferation, and bone ingrowth are enhanced [19].

## Figures and Tables

**Figure 1 materials-16-02484-f001:**
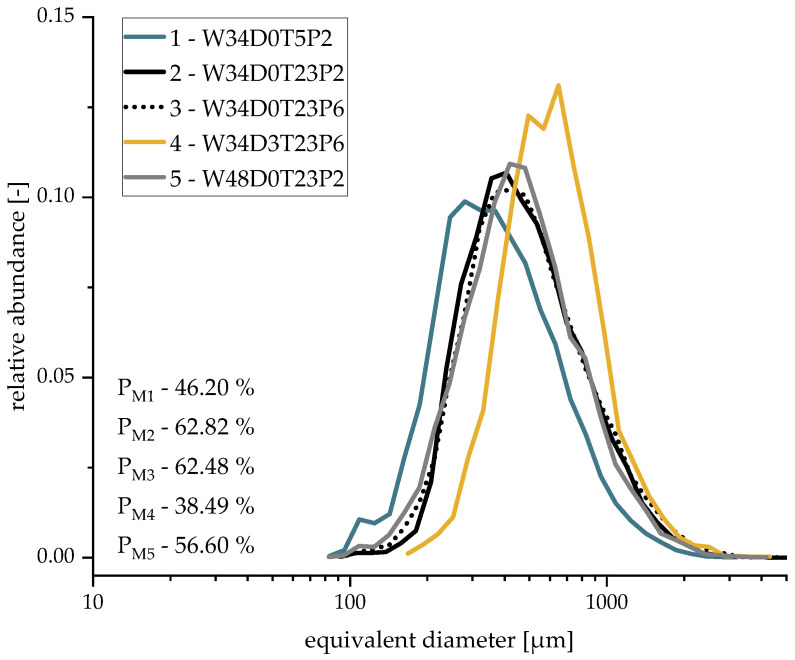
Pore size distribution of macroscopic pores (via X-ray).

**Figure 2 materials-16-02484-f002:**
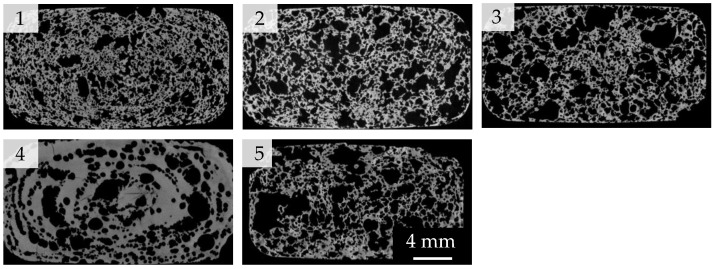
Structure of sintered Freeze Foams (1-W34D0T5P2; 2-W34D0T23P2; 3-W34D0T23P6; 4-W34D3T23P6; 5-W48D0T23P2) via X-ray.

**Figure 3 materials-16-02484-f003:**
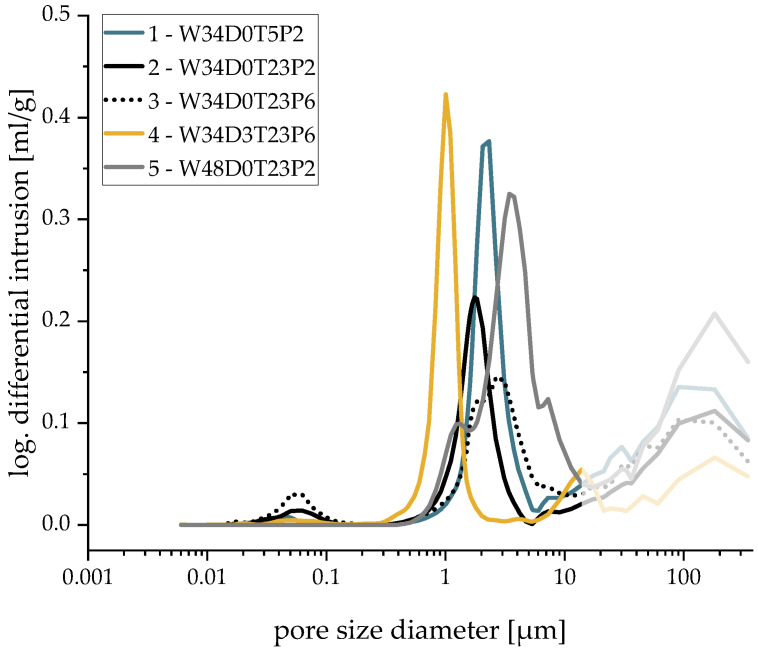
Mercury porosimetry results—logarithm differential intrusion of mercury.

**Figure 4 materials-16-02484-f004:**
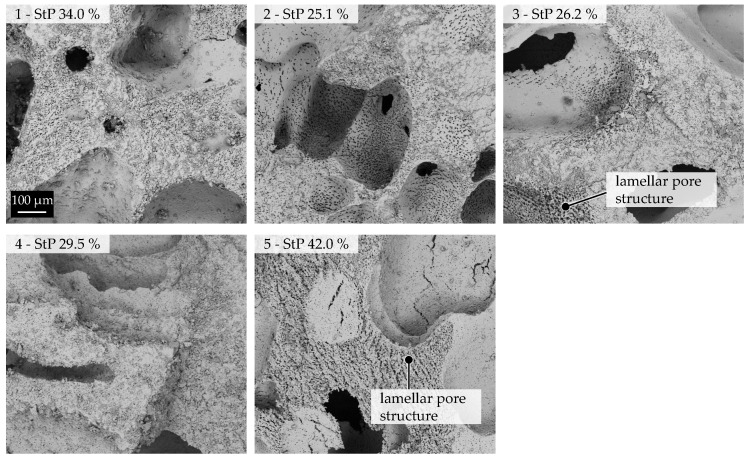
Polished cross-sections at 100× magnification recorded by FESEM.

**Figure 5 materials-16-02484-f005:**
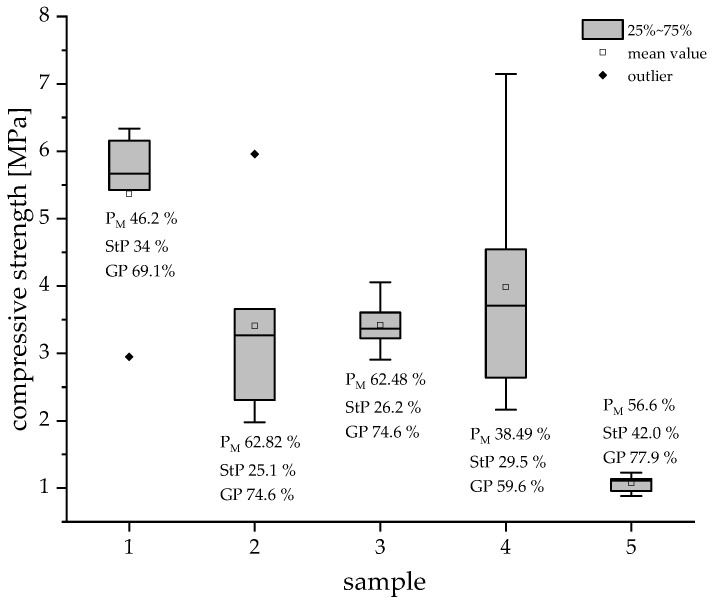
Compressive strengths of Freeze Foam specimen including mean value, outliers, and standard deviation.

**Figure 6 materials-16-02484-f006:**
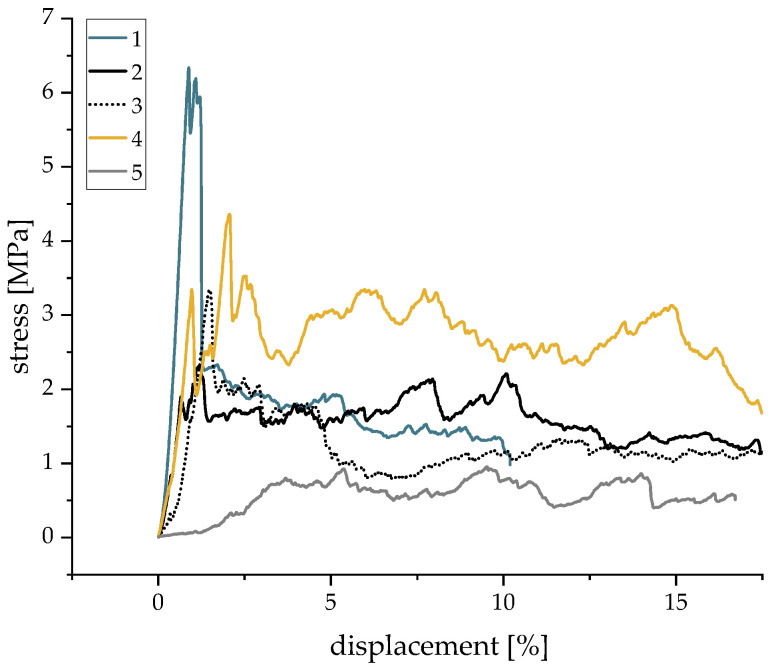
Stress–displacement curves for a representative sample of each configuration, 1–5.

**Figure 7 materials-16-02484-f007:**
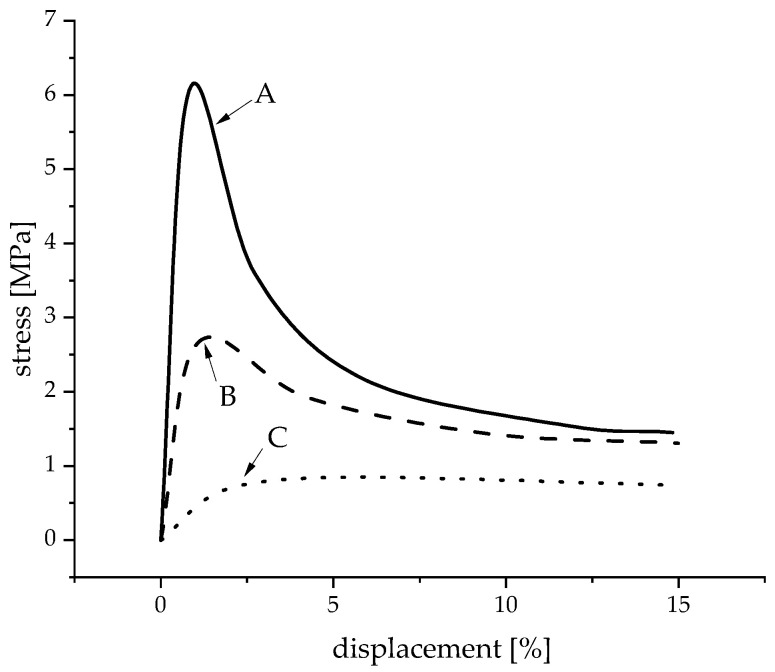
Schematic stress–displacement curves (modes of failure: A—high force maximum; B—medium force maximum; C—constant, low force maximum).

**Figure 8 materials-16-02484-f008:**
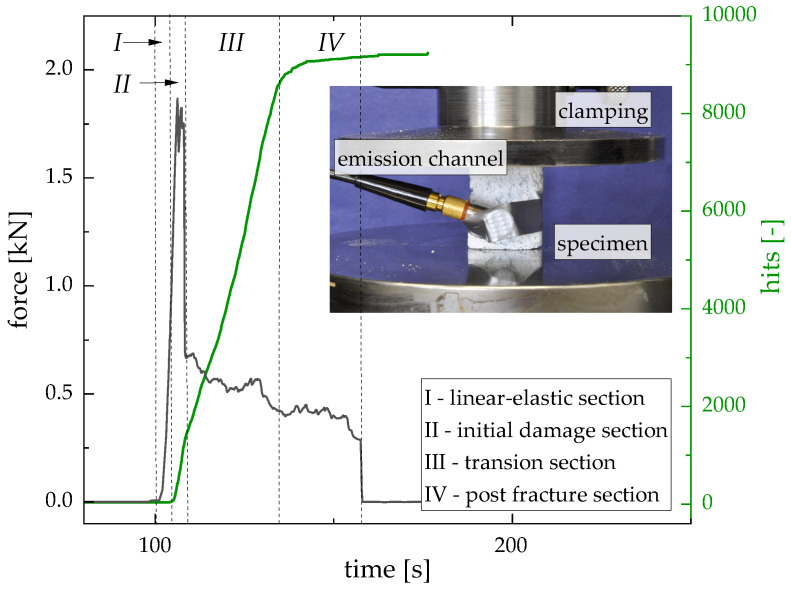
Correlation of force–time curve with acoustic emission analysis.

**Table 1 materials-16-02484-t001:** Parameter configuration of the investigated freeze foams.

Sample	W (m.-%)	D (min)	T (°C)	P (min)
1-W34D0T5P2	34	0	5	2
2-W34D0T23P2	34	0	23	2
3-W34D0T23P6	34	0	23	6
4-W34D3T23P6	34	3	23	6
5-W48D0T23P2	48	0	23	2

**Table 2 materials-16-02484-t002:** Mean values of the pore sizes from microscopic (via mercury porosimetry) and macroscopic pores (via X-ray).

Sample	Peak 1 (µm)	Peak 2.1 (µm)	Peak 2.2 (µm)	Peak 3 (µm)
1	micro-cracks	2.0	-	280
2	micro-cracks	1.8	-	406
3	micro-cracks	1.8	3.1	418
4	micro-cracks	1.0	-	648
5	-	1.2	3.4	479

## Data Availability

Not applicable.

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
