# Peer review of "Investigation of the Damage Phenomenology with Dependence on the Macroporosity and Microporosity of Porous Freeze Foams"

_materials, 2023, doi:10.3390/ma16062484_

Round 1

Reviewer 1 Report

Revision of “Investigation of the Damage Phenomenology in Dependence on the Macro- and Microporosity of porous Freeze Foams”

The manuscript under review devoted to experimental study the correlation between the pore forming influencing factors and the resulting mechanical properties in Freeze Foams. Providing of such investigations is very important from an academic and economic points of view.

The following three failure types of structural stability were highlighted by authors identified whose occurrence depends on the micro and macro porosities exhibited by the different samples

A: High and well-defined force maximum due to partly closed macroscopic pores with low porosity and closed microscopic pores that was observed mostly in samples with W34 and T5.

B: Medium force maximum because of interconnected macroscopic porosity with a high total porosity and closed and low number of micro pores that was a result of elevated temperatures (T23) and low water content.

C: Almost constant, low resistance to deformation due to interconnected macroscopic and microscopic pores and high porosities that was caused by high water content (W48) and the resulting lamellar pores due to freezing.

Acoustic emission analysis allowed for a description of damage phenomenology and the identification of four damaging phases.

In manuscript all necessary information is captured by 8 figures and 2 tables. There are 19 references.

After getting acquainted with the presented manuscript, a few small questions remained:

1.      It is necessary to give an explanation, if it possible, namely, whether it is possible to extend the obtained results to other initial components, since section 2 indicates what was taken as a basis. For example, suspension hydroxyapatite, etc.

2.      Were other technological regimes used in obtaining the objects of study. And if not, why not?

3.      Is it possible to speak about the reliability of the results presented in Fig. 5 for sample 4 if there is such a spread of the standard deviation?

Obtained results are important both for understanding the physical processes that occur in real objects and for the development of new devices. It corresponds to the field of the Journal «Materials». It may be accepted after small revision.

Author Response

Dear Reviewer, 

attached you will find the document with the revisions and respones we made.

Best Regards,

Johanna Maier

Reviewer 2 Report

Comments from Reviewer

Title: Investigation of the Damage Phenomenology in Dependence on the Macro- and Microporosity of porous Freeze Foams

The current form's presentation of methods and scientific results is unsatisfactory for publication in the Materials journal. The minor and significant drawbacks to be addressed can be specified as follows:
1.    Title. porous – Porous.
2.    Do not repeat the words in the title to the “keywords”.
3.    Lines 33-56. this paragraph is too long. authors should clearly state the purpose of their research and very briefly write what they will research and how. This is not an abstract and conclusion!!!!
4.    Lines 112 and 113.
5.    Too short a paragraph.
6.    Tab. 1, lines 114 – 119, and Fig. 1. the authors use different names for the samples. This makes it challenging to analyse the data. I suggest entering the full names of the samples in tab. 1.
7.    Line 114. M2? M3?
8.    Fig. 1, y-axis. rel. ---> relative.
9.    Fig. 1. were these results obtained based on mercury porosimetry?
10.    Fig. 2 (figure captions) and line 78. CT: computed tomography? or compact tomography?
11.    Fig. 3, figure captions. Log?
12.    Figs. 1 and 3 – see legends. Please unify the naming of the samples. Why did the authors not use the same markings for individual samples?
13.    Fig. 3, y-axis. (i) differential? (ii) Log Differantial Intrusion ---> log differential intrusion.
14.    Tabs. 1 and 2. Number? Or Sample?
15.    Lines 82. FESEM? See line 162.
16.    Where do the authors show the microporosity results? What measurement technique did they use?
17.    Fig. 7. please explain in the figure captions what explain what A, B, and C mean. What is the relationship A, B, and C with A, B, and C in the lines 257 - 265?
18.    Fig. 8. Not a very fortunate use of the word "area".
19.    Conclusions?
20.    Literature should also be standardized: the size of letters in the titles of journals, and initials of names, the size of letters in the titles of articles. For examples, see and compare [17] and [18].

Sincerely
     The reviewer.

Author Response

Dear Reviewer, 

attached you will find the document with the revisions and responses we made.

Best Regards,

Johanna Maier

Round 2

Reviewer 2 Report

The authors have made a substantial improvement in this article. The manuscript can be accepted for publishment in the present form. Overall the manuscript improved. I have two questions:

1.      Fig. 3. I doubt whether the values on the y-axis are logarithmic.

2.      Does mercury allow you to explore micropores? No. See, for example, http://www.mcaservices.co.uk/pore%20size%20distribution.htm

https://wiki.anton-paar.com/en/mercury-intrusion-porosimetry-basics-measuring-pores-in-solids/

Author Response

Dear Reviewer,

attached you will find the comments to your review.

Best Regards.

Johanna Maier
